# Commensal *Neisseria* Are Shared between Sexual Partners: Implications for Gonococcal and Meningococcal Antimicrobial Resistance

**DOI:** 10.3390/pathogens9030228

**Published:** 2020-03-19

**Authors:** Christophe Van Dijck, Jolein G. E. Laumen, Sheeba S. Manoharan-Basil, Chris Kenyon

**Affiliations:** 1Department of Clinical Sciences, Institute of Tropical Medicine Antwerp, 2000 Antwerp, Belgium; 2Department of Medicine, University of Cape Town, Cape Town 7700, South Africa

**Keywords:** commensal, *Neisseria*, gonorrhea, *meningitidis*, kissing, sharing, microbiome, transmission, antimicrobial resistance

## Abstract

Antimicrobial resistance in pathogenic *Neisseria* parallels reduced antimicrobial susceptibility in commensal *Neisseria* in certain populations, like men who have sex with men (MSM). Although this reduced susceptibility can be a consequence of frequent antimicrobial exposure at the individual level, we hypothesized that commensal *Neisseria* are transmitted between sexual partners. We used data from a 2014 microbiome study in which saliva and tongue swabs were taken from 21 couples (42 individuals). Samples were analyzed using 16S rRNA gene sequencing. We compared intimate partners with unrelated individuals and found that the oral *Neisseria* communities of intimate partners were more similar than those of unrelated individuals (average Morisita–Horn dissimilarity index for saliva samples: 0.54 versus 0.71, respectively (*p* = 0.005); and for tongue swabs: 0.42 versus 0.63, respectively (*p* = 0.006)). This similarity presumably results from transmission of oral *Neisseria* through intimate kissing. This finding suggests that intensive gonorrhea screening in MSM may, via increased antimicrobial exposure, promote, rather than prevent, the emergence and spread of antimicrobial resistance in *Neisseria*. Non-antibiotic strategies such as vaccines and oral antiseptics could prove more sustainable options to reduce gonococcal prevalence.

## 1. Introduction

*Neisseria gonorrhoeae* has rapidly acquired resistance to all antimicrobials used to treat it, and there is a real risk that it may be untreatable in the near future [1]. It is increasingly appreciated that a key way it acquires this antimicrobial resistance (AMR) is via taking up resistance genes from oropharyngeal commensal *Neisseria*. The genus *Neisseria* is one of the three most abundant phyla in the human oral microbiome [2], with almost all individuals being colonized with at least one *Neisseria* species [3]. This high prevalence, in combination with extensive antimicrobial exposure, is thought to explain the extensive AMR in commensal *Neisseria* that has been found in certain populations, like cohorts of men who have sex with men (MSM) [4] and that has played an important role in the genesis of AMR in *N. gonorrhoeae* [5].

Epidemiological and modeling studies evaluating the emergence of AMR in *N. gonorrhoeae* have typically included the sexual transmission of resistant gonococci but not commensal *Neisseria* [6,7]. If resistant commensal *Neisseria* were also sexually transmitted, this would be important to take into consideration. This would be particularly important if these commensals could be transferred via highly prevalent activities such as tongue kissing. Transfer via kissing would diminish the likelihood that traditional gonorrhea control measures would work to control the genesis and spread of gonococcal AMR. In certain instances, they may even be counterproductive. Several authors have, for example, suggested that because pharyngeal gonorrhea plays such an important role in the emergence of AMR (via horizontal gene transfer from commensals), intensive screening and treatment of pharyngeal gonorrhea in MSM should be advocated [1]. This strategy has been shown to result in extremely high antimicrobial exposure with a resultant high probability of inducing AMR in commensal *Neisseria* [8]. If these resistant *Neisseria* were then transferred via kissing and these resulted in AMR in *N. gonorrhoeae*, then intensive screening may indirectly increase rather than decrease the probability of gonococcal AMR emergence.

Concerns around the transmission of commensal *Neisseria* via kissing have emerged following increasing evidence of this mode of transmission for related bacteria. Several studies have found that kissing is a risk factor for meningococcal disease [9,10,11] or carriage [12,13,14,15] among students. Likewise, *N. gonorrhoeae* can be readily cultured from saliva [16,17,18], saliva use as a lubricant is a risk factor for rectal gonorrhea [19], kissing [20,21,22] as well as having a main partner with pharyngeal gonorrhea [23] may be risk factors for pharyngeal gonorrhea and a mathematical transmission model showed that oro–oral transmission is essential to generate the actual prevalence of gonorrhea among MSM [6].

Furthermore, a number of studies have found that the oral microbiome is shared between household members [24,25]. An important study by Kort et al. in 2014 demonstrated that intimate partners share a similar oral microbiome and that the degree of similarity of the salivary microbiota correlates with the kissing-frequency in the past weeks and with the time since the last kiss [26]. They calculated that an intimate kiss of 10 seconds leads to an average transfer of $10^{8}$ bacteria from one partner to another [26].

These considerations led us to hypothesize that commensal *Neisseria* are transmitted between sexual partners. To test this hypothesis, we performed a secondary analysis of the study by Kort et al. We found that kissing partners shared more similar *Neisseria* communities than unrelated individuals.

## 2. Results

The dataset provided by Kort et al. [26] consisted of tongue and salivary microbiota samples taken from 21 couples visiting a Zoo in 2012. We compared the results from the entire range of 3000 operational taxonomic units (OTUs) with those from the 66 OTUs which represent members of the genus *Neisseria*. We found that pairwise comparison of samples using the Morisita–Horn dissimilarity index (MHi) did not differ significantly for analyses based on the entire versus the restricted dataset. Based on *Neisseria*-related OTUs we found the following:A high pairwise similarity (an MHi value close to zero) between duplicate samples of an individual’s tongue surface (MHi 0.17) and saliva (MHi 0.28) indicated that sampling was reproducible at the level of the genus *Neisseria* (Figure 1).Partners’ oral *Neisseria* communities sampled after a 10-second kiss were not more similar than before the kiss (saliva: average MHi 0.55 before versus 0.53 after, *p* = 0.704; surface of the tongue: average MHi 0.39 before versus 0.45 after, *p* = 0.597; Figure 1). Therefore, samples before and after kissing were combined in the subsequent analyses.Partners’ oral *Neisseria* communities were more similar compared to unrelated individuals. This was found for saliva (average MHi 0.54 versus 0.71, respectively, *p* = 0.005) and for samples of the tongue surface (average MHi 0.42 versus 0.63, respectively, *p* = 0.006; Figure 1).

## 3. Discussion

Although it was already known that household members and intimate partners share oral commensal microbiota [24,25,26], the current analysis demonstrates that intimate partners also share similar commensal *Neisseria*. This is a logical, yet important finding, as commensal *Neisseria* are known to harbor several AMR determinants [27] that are a frequent source of AMR for pathogenic *Neisseria* [4,28,29].

Sharing of commensal *Neisseria* via this and other modalities may, therefore, explain the high prevalence of antimicrobial resistant commensal *Neisseria* in certain groups of patients. A study from Japan in 2005–2006, reported the antimicrobial susceptibility of 45 oropharyngeal *Neisseria subflava* isolates from men with urethritis and female commercial sex workers. The majority of isolates had reduced susceptibility to penicillin, tetracycline and ciprofloxacin [30]. Another study in Vietnam in 2016–2017 investigated 265 *Neisseria* isolates from 207 MSM, including 9 gonococci and 13 meningococci. Ten different *Neisseria* species were identified. Twenty-eight percent of samples had reduced susceptibility to ceftriaxone (minimum inhibitory concentration ≥0.125 mg/L) [4]. The reason for the high prevalence of commensal *Neisseria* with reduced antimicrobial susceptibility in these groups of patients presumably parallels the one proposed for gonorrhea: repeated cycles of reinfection/recolonization and antimicrobial exposure in individuals within a highly connected transmission-network [31].

In addition, since the pharynx is the predominant reservoir of nonpathogenic *Neisseria* in humans, it is probable that *Neisseria* are transmitted between partners by transfer of saliva, either directly (by intimate kissing or through aerosolized droplets), or indirectly (e.g., through shared fomites). The scarcity of nonpathogenic *Neisseria* within other bodily niches makes it unlikely that the skin, genital or anorectal site act as an intermediate in this transfer process. As already noted, different types of evidence suggest that pathogenic *Neisseria* species can be transmitted by kissing [6,9,10,11,12,13,14,15,16,17,18,19,20,21,22,23]. Our findings support to the idea that the genus *Neisseria* can be transmitted by kissing.

The limitations of this study include the following. First, the fact that partners share certain microbiota does not provide direct evidence of transmission between them. Intimate kissing may be one explanation, but we have not explored alternative means of transmission. Potential mediators of transmission could be via fomites or animals (such as pets), or influences on the oral microbiota by environmental factors, common diet or simultaneous exposure to pathogens, toxins, mouthwashes or antimicrobials [32]. Second, identification of the oral microbiota in this study was based on the amplification of hypervariable regions V5–V7 of the 16S rRNA gene. This does not allow for the accurate identification of microbiota at the species level, nor does it provide information concerning antimicrobial susceptibility of the microbiota involved. Still, it seems reasonable to infer that sharing of specific OTUs represents sharing of a specific subset of bacterial genomes and, thus, AMR determinants within these bacteria.

The significance of this study lies in its relevance for preventing the further emergence of AMR in *N. gonorrhoeae* and *N. meningitidis*. If commensal *Neisseria* can be spread by common-place activities such as kissing, then this increases the probability that intensive gonorrhea screening in high prevalence populations such as MSM will, via increased antimicrobial exposure, promote, rather than retard, the emergence of AMR in *Neisseria*. Certain groups of at-risk populations are frequently exposed to antibiotics to treat symptomatic sexually transmitted infections. Treatment of asymptomatic cases increases this exposure even more. As most cases of anorectal and pharyngeal gonorrhea are asymptomatic, regular screening of asymptomatic patients results in a much higher number of diagnosed infections and, thus, a substantial increase in antibiotic exposure [33]. Currently, several guidelines recommend regular gonorrhea screening among MSM at high risk of infection [34,35]. The idea behind this is that treatment of all cases of gonorrhea in a population would eventually lead to a reduction (or eradication) of the pathogen from that population. There is, however, very little empirical evidence that supports this hypothesis [36]. On the other hand, increased antimicrobial exposure has been linked to AMR in gonorrhea [37,38]. This, together with the finding from the current study that *Neisseria* (including AMR determinants) may be transmitted to other individuals within a network via kissing, provides another pathway for the dissemination of AMR. Intensive screening and treatment of all positives may have a profound impact on the prevalence of AMR in commensal *Neisseria*, which could then be rapidly spread between individuals by kissing. A more prudent approach to preventing the emergence of AMR would be to reduce antimicrobial exposure as far as possible. This could include reduced screening and using non-antibiotic strategies such as vaccines and oral antiseptics to reduce gonococcal prevalence [39,40].

## 4. Materials and Methods

### 4.1. Sample Collection and Processing

In the study by Kort et al., samples were collected from 42 individuals (21 couples) visiting a Zoo in the Netherlands in 2014. A swab was taken from the anterior dorsal surface of the tongue and saliva was collected in a sterile 15 mL tube. Each participant was sampled before and after an intimate kiss of 10 s. Three couples were sampled in duplicate in order to assess reproducibility. Samples were stored at −80 °C until further processing. After DNA extraction, quantitative 16S rRNA PCR was used to generate an amplicon library based on the 16S variable regions V5-V7. Aligned 16S rRNA sequences were clustered into OTUs, defined by 97% sequence similarity. The RDP Naive Bayesian Classifier and the SILVA reference database (release 119) were used for taxonomic classification. The full study protocol is described in the original paper [26].

### 4.2. Availability of Data and Materials

The dataset supporting the conclusions of this article is available as a supplementary file to the paper by Kort et al. [26] For the *Neisseria*-specific analysis, the dataset was restricted to only those 66 OTUs representing members of the genus *Neisseria*.

### 4.3. Assessment of Community Similarity

Similarity of tongue and salivary microbiota (β-diversity) was determined by calculating pairwise distances with the Morisita–Horn dissimilarity index [41] using R version 3.6.1. A value of zero on this index represents complete similarity, whereas a value of one means complete dissimilarity.

### 4.4. Statistical Analysis

The non-parametric Wilcoxon rank-sum test in R was used to calculate the *p*-values for selected paired differences of data. Data were visualized using Microsoft Excel.

### 4.5. Ethics Approval and Consent to Participate

Not applicable.

## Figures and Tables

**Figure 1 pathogens-09-00228-f001:**
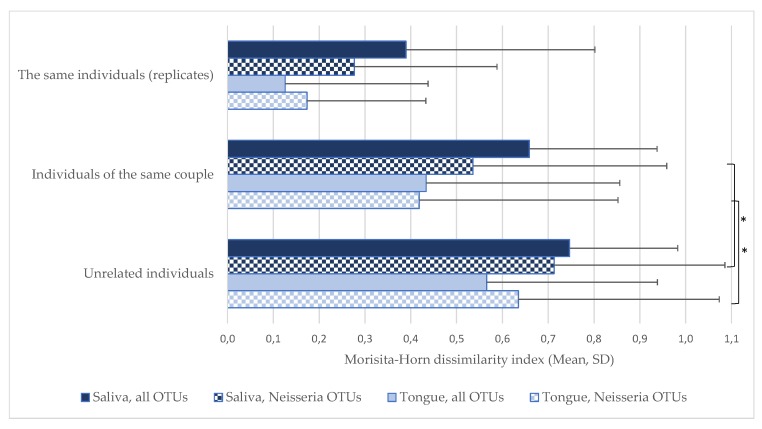
Morisita–Horn dissimilarity indices of samples from the same individuals, intimate partners and unrelated individuals. An index of 0 represents complete similarity whereas an index of 1 means complete dissimilarity. Each bar shows the average Morisita–Horn index, whiskers indicate standard deviations, * *p* < 0.01.

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
