# Peer review of "Commensal Neisseria Are Shared between Sexual Partners: Implications for Gonococcal and Meningococcal Antimicrobial Resistance"

_pathogens, 2020, doi:10.3390/pathogens9030228_

Round 1

Reviewer 1 Report

Dear Authors,

     I have reviewed the manuscript entitled "Commensal Neisseria are shared between sexual partners: implications for gonococcal and meningococcal antimicrobial resistance" and found it to be very well written, organised and of interest for the scientific community.

Author Response

We are very thankful to the reviewer for his/her positive response on the manuscript.

Reviewer 2 Report

The manusciprt describe the possibility that commensal Neisseria plays a role in spreading AMR through intimate action such as kissing. Therefore, the antimicrobial treatment may increase AMR not only on the patient but also the partner. Overall the story makes sense and it is what we need to pay attention on. However, I have some questions/suggestions. 

Line 11-12: I actually do not understand how screening would affect the spread of AMR. Shouldn't it be antimicrobial treatment?

Line 6: 16srRNA gene sequencing.

Figure 1: I would have stars on the figure so I know the significant difference. 

It would be better if a PCoA/NMDS figure of these Neisseria species to show partner based grouping.

I would have another figure showing top few Neisseria species that mostly fit into your figure 1 so we know that those species are targeted AMR spreading species.  

Like the author said, microbiota from saliva and tongue may change a lot in a short time due to diet, mouthwash, diet. And we really don't know if the partners sharing the same microbiota is the result of kissing or just the same living habit? How one time sampling can show an usually long term AMR spreading is very hard. But you may want to give more clues on future targeted Neiserria species (top three from your data) so you can screen on the AMR genes from them. 
